# Research on the Effect of Rural Inclusive Financial Ecological Environment on Rural Household Income in China

**DOI:** 10.3390/ijerph19042486

**Published:** 2022-02-21

**Authors:** Heping Ge, Lianzhen Tang, Xiaojun Zhou, Decai Tang, Valentina Boamah

**Affiliations:** 1School of Management Science and Engineering, Nanjing University of Information Science & Technology, Nanjing 210044, China; 002251@nuist.edu.cn (H.G.); 20201242035@nuist.edu.cn (L.T.); 20215242005@nuist.edu.cn (V.B.); 2School of Business, Jiangsu Open University, Nanjing 210005, China; 3China Institute of Manufacturing Development, Nanjing University of Information Science & Technology, Nanjing 210044, China

**Keywords:** digital inclusive finance, rural residents, income increase effect, spatial effect

## Abstract

After a long struggle against poverty, the problem of absolute poverty among Chinese rural residents has been solved, but the problem of relative poverty still exists. With digitalization, the ecological environment of rural inclusive finance has been optimized. This paper empirically tests the individual fixed-effect model and finds that digital inclusive finance has a positive income-increasing effect on rural residents. Wage income, operating income, and transfer income among the income types undergo a certain degree of promotion, while property income is affected to the contrary. In addition, digital inclusive finance has the same effect on farmers’ income increases in the east and central regions of China. However, it has a slightly smaller impact on farmers in the west. This paper uses a spatial econometric model and finds that promoting the development of local digital inclusive finance will enhance the income level of local farmers and increase the income of neighboring farmers. Therefore, this paper proposes to speed up the development of digital inclusive finance, optimize the rural financial ecological environment, strengthen government supervision and other recommendations, further enhance farmers’ income, and achieve common prosperity.

## 1. Introduction

The widening of poverty and the income gap has become one of the serious challenges facing the development of the world today, attracting the international community’s attention, especially the developing countries [1,2]. On 25 February 2021, the Chinese government officially declared an all-out victory in China’s fight against poverty, marking the country’s completion of the historic task of eradicating absolute poverty. The Chinese government has made a series of remarkable achievements since 2013, according to the National Bureau of Statistics of the People’s Republic of China. Firstly, the total income of all residents has been increasing continuously, and the per capita disposable income has increased from 18,300 yuan in 2013 to 32,200 yuan in 2020. Secondly, the income gap between urban and rural areas has gradually narrowed, with the ratio of urban residents to rural residents per capita disposable income dropping from 2.81 times in 2013 to 2.56 times in 2020. Thirdly, the source structure of residents’ income has become more stable and diversified. The share of residents’ property income and transfer income has increased from 2% and 17.5% in 2013 to 2.5% and 21.4%, respectively, in 2020.

However, the ending of poverty is not the end but a new starting point. After eliminating absolute poverty, the problem of relative poverty in China remains severe and will remain for a long time to come. How to consolidate the achievements of poverty eradication and increase the income of rural residents has become an important issue facing the all-round construction of a great modern socialist country [3]. A good rural financial ecological environment can promote the efficient operation of rural finance. Research by Mahjabeen has found that inclusive financial development can significantly boost the incomes of all segments of society, especially rural dwellers [4]. In September 2016, China proposed at the G20 Hangzhou summit the formulation of the G20 senior principles on digital inclusive finance, advocating digital technologies to support the development of inclusive finance and construction of digital financial infrastructure ecosystem. As a new form of inclusive finance, digital inclusive finance has the advantages of low cost, wide coverage, and more convenient and fast financial services. This paper focuses on answering the following questions: can the development of digital inclusive finance further boost the income of rural residents? In what form will its growth be boosted? Is there any difference in farmers’ income in different regions? Exploring these questions has important practical significance under the background of the New Era in China. China is the largest developing country globally, and the regional development is quite different. Studying the revenue-increasing effect of Chinese digital inclusive finance has important reference significance for policy makers in designing and implementing poverty reduction strategies, reducing the incidence of poverty and income inequality [5].

## 2. Literature Review

In 1935, the British ecologist Tessler first put forward the concept of the ecosystem. In the following decades, financial scholars have been trying to develop the theory of the financial ecosystem. The term “financial ecology” comes from “optimizing financial ecology” put forward by Xiaochuan Zhou in 2004; the two basic concepts of financial ecological subject and financial ecological environment have evolved gradually. Recently, the new theory of the financial ecosystem has become more and more refined. The research on the financial ecological environment mainly includes three aspects: the influencing factors in financial efficiency, the relationship between financial efficiency and financial ecological environment, and the countermeasures to the perfect financial ecological environment. To improve rural financial institutions’ micro-operation efficiency, Paul proposed establishing rules with an effective supervision system for reliable and fair implementation [6]. Andrew Lawson believes that the unstable macroeconomic environment, the biased policies of cities, the rigidity of financial markets, and political and legal constraints are important factors affecting the efficiency of financial operations [7]. Allen proposed to measure the efficiency of rural financial institutions in terms of the policy environment, legal regulation control system, market barriers, incomplete information and communication, and the effectiveness of government intervention [8].

A good financial ecological environment can promote the healthy and efficient development of the financial system [9], thus attracting capital accumulation and reducing operating costs [10]. King and Levine, who built an evaluation index system and a rural credit risk early-warning model based on their research on rural financial development, rural economic development, and the rural credit legal environment, found that rural credit information asymmetry is the main factor leading to rural credit risk [11]. Establishing a sound financial ecological environment assessment system is conducive to the rational and effective allocation of regional financial resources. The diversification of financial institutions is the basic element to maintaining financial vitality. Claessens et al. found that the development of financial institutions and the diversification of financing channels play a positive role in reducing rural poverty, solving the imbalance between urban and rural development levels, and promoting rural economic development [12] and rural financial development can increase farmers’ income [13]. Blowers empirically assessed the rural financial ecology of two large developing countries, India and China, and found that rural economic communities largely depend on informal financial institutions. The main reasons for the existence of informal financial institutions are the institutional defects of formal financial organizations, incomplete construction of credit systems, and limited support capacity of national agricultural policy [14]. In addition, financial support from the government will help raise the income of rural residents [15]. Exchange rates, gross domestic product, and technology can boost rural incomes, but interest rates and rural land prices are inversely related to rural incomes [16].

Inclusive finance is different from the traditional financial model. It is a top-down innovative financial development model. Its basic goal is to ensure equitable and inclusive growth; inclusive finance helps promote personal financial stability and extends the reach of traditional financial services by lowering barriers to financial services and easing financial exclusion [17,18], providing long-tail customers with more equitable and inclusive financial services to achieve poverty reduction and revenue increase effect [19,20]. Kapoor believes inclusive finance can accompany economic development, promoting urban–rural co-development [21]. Shen argues that inclusive finance provides fair credit rights and access to finance for all, and reduces the economic risk for poor households [22]. Through the study of developing countries in Asia, Park et al. found that inclusive financial development can promote the growth of per capita GDP [23]. Beck found that inclusive finance helps alleviate the financing problems of small and medium-sized enterprises, deliver economic vitality, and promote economic growth [24]. Bittencourt focused on low-income groups in Brazil, and concluded that expanding the reach and efficiency of financial services would raise the standard of living of about 20 percent of the poor [25]. Sarma’s study found a positive correlation between inclusive finance and gross national product [26]. Sahay et al. calculate the level of financial development in inclusive finance from an enterprise and personal perspective in 88 countries, and conclude that inclusive financial development has a long-term relationship with economic growth [27]. Corrado studies microeconomic and macroeconomic perspectives, and believes that promoting inclusive financial business can contribute to the local economy and that inclusive financial contribution to economic growth is long-term sustainable [28]. Hannig and Jansen argue that inclusive finance should expand the scope of its financial services by bringing users excluded from the banking system into the formal financial system and increasing the availability of financial resources [29].

Inclusive finance can effectively raise the average income level of rural households [30]. Using India as an example, Mor and Ananth argue that establishing an inclusive financial system can effectively reduce the cost of financial services, reduce the national income gap, increase the growth rate of the economy, effectively reduce poverty levels in rural areas, and promote social development [31]. Liu et al., based on the survey data of 988 rural households in China’s poor counties, found that the development of inclusive finance has promoted the rural inclusive transformation of entrepreneurial activities of rural households in China’s rural areas, thus helping to raise the income of rural residents and potentially narrowing the income gap [32].

However, there have been some shortcomings in the development of inclusive finance [33]. The low penetration rate of digital technology in rural areas makes rural residents less motivated to access digital inclusive financial services [19]. Diniz believes inclusive financial development will widen the income gap between urban and rural areas because rural residents lack financial knowledge [34].

Since the popularity of the internet and mobile devices has increased, China has vigorously promoted the integration of inclusive finance and digital technology and pioneered the concept of “Digital Inclusive Finance”, new types of financial services to supplement traditional financial services. Peterson’s study suggests that digital finance can promote inclusive finance in developing and developed countries. Further, low-income and non-fixed-income people can obtain financial services from financial institutions at lower costs [35]. Information markets in developing countries are relatively inefficient compared with those in developed countries. “Digital + inclusive finance” has the characteristics of lowering the threshold of financial services, promoting price discovery and information circulation, and opening the “last mile” of financial services. It is a mode of realizing high financial coverage, low cost, and sustainable development. At the same time, digital inclusive finance has gained international recognition for its pro-poor nature.

Since 2010, the G20 and the World Bank have led initiatives to increase financial inclusion in developing countries to help reduce poverty levels in developing countries and emerging economies. The development of digital inclusive finance has made it possible for more rural residents to obtain basic financial services, such as loans, savings, mobile payments, and insurance at an affordable cost to achieve income growth. Galak argues that digital finance will allow a rapid expansion of financial services to reach more people [36]. Gabor and Brooks propose that low-income groups will rely on the combination of traditional finance, digital technology, and higher financial accessibility [37]. Asongu and Nwachukwu studied the relevance of mobile banking use on poverty reduction and income inequality in developing countries using cross-sectional data from 93 developing countries in 2011 relevance [38]. Jie et al. based on China panel data, studied how household consumption was affected by digital finance, and finally proposed that transactions and payments could be facilitated by the development of digital finance. The consequent expansion of investment channels, increase in income, and increase in household consumption, especially in the third-and fourth-tier cities and in households with relatively low income and fewer assets, would have a very significant growth effect, due to the inclusive nature of digital finance [39].

By combing the previous relevant literature, we find that optimizing the financial ecological environment and the development of inclusive finance and digital inclusive finance positively affect farmers’ incomes. With the support of digital technology, the coverage of rural financial services has been improved, making up for the lack of inclusive finance. However, many scholars put forward this view based more on the inclusive financial perspective, to analyze rural financial services that need digital technology to solve the problems of traditional inclusive finance. From the digital point of view, research on the current rural finance situation is still relatively negligible. Therefore, this paper focuses on the impact of digital inclusive finance on rural income to explore whether the development of digital inclusive finance can further promote rural income growth, and in what form it promotes this growth. In this paper, the research contents are divided as follows: first, we study the effect of the digital inclusive finance aggregate index on farmers’ incomes. Second, we study the effect of six sub-indicators on farmers’ incomes. Third, we divide farmers’ incomes into four categories and then study the influence of the total index of digital inclusive finance on their sub-incomes. Finally, the spatial metrological model is used for further analysis.

## 3. Model Construction and Data Explanation

### 3.1. Theoretical Mechanism

With the advent of the digital age, big data analysis and cloud computing functions have been developed rapidly. The integration of digital technology into inclusive finance has further deepened inclusive finance, expanded the range of available financial services, and reached more vulnerable groups. The problem of financial exclusion faced by rural residents has been effectively alleviated. At the same time, the development of digital inclusive finance provides rural residents with low-cost, low-threshold insurance and credit support [40]. It enables rural residents to innovate and start businesses, thus increasing their income channels. Therefore, based on the above analysis, Hypothesis 1 is proposed.

**Hypothesis** **1** **(H1).**
*The development of digital inclusive finance can increase the income of rural residents.*


Although the degree of economic development varies in different regions of our country, digital inclusive finance has broken through regional boundaries with deeper degrees of digitalization. This has made it easier for rural residents and poor areas previously excluded from traditional financial institutions. Digital inclusive finance can promote equal opportunity, contribute to inclusive social growth and rational allocation of resources, and reduce the urban–rural income gap. Therefore, based on the above analysis, Hypothesis 2 is proposed.

**Hypothesis** **2** **(H2).**
*Digital inclusive finance has a positive income-increasing effect on rural residents in China’s east, middle, and west.*


With the development of the internet and with e-commerce platforms booming, cross-regional transactions have increased. On the one hand, digital inclusive finance uses big data processing technology to reduce market information asymmetry, reduce transaction costs of rural e-commerce transactions, and provide residents with more convenient payment services; on the other hand, it lowers the threshold of credit so that more rural residents can participate in e-commerce sales and ease the problem of the backlog of unsalable agricultural products. In addition, the development of digital inclusive finance has brought income to rural residents in the same region, which has roughly the same types of crops due to geographical and weather factors. At the same time, it can also promote the economic development of the surrounding areas. Based on the above analysis, Hypothesis 3 is proposed.

**Hypothesis** **3** **(H3).**
*The revenue-raising effect of digital inclusive is spatially spillover.*


### 3.2. Modeling

In order to test Hypotheses 1 and 2 and study the effect of digital inclusive finance on the income of rural residents, the following individual fixed effect model was constructed:(1)$$DI_{it}=\alpha_{0}+\alpha_{1}DIFI_{it}+Open_{it}+Edu_{it}+Urban_{it}+Govern_{it}+Traffic_{it}+\psi_{i}+\varepsilon_{it}$$

In Formula (1), “*i*” is the province, and “*t*” is the year. $DI_{it}$ is the disposable income per capita for rural residents, the explanatory variable for this paper, it consists of four parts: average wage income (WI), average operating income (OI), average property income (PI), and average transfer income (TI). $DIFI_{it}$ is the digital inclusive financial index, the core explanatory variable of this paper, and has selected six representative sub-variables in its composition for detailed analysis. These include DCB, DUD, DSS, Payment, Insurance, and Credit. This paper selects the following five control variables: $Open_{it}$ means the level of opening to the outside world, expressed in each region’s total imports and exports as a percentage of GDP. $Edu_{it}$ means educational level, expressed by the average number of years of education of employed persons in each region. $Urban_{it}$ represents the level of urbanization, expressed in terms of the proportion of the urban population in the resident population of each region. $Govern_{it}$ is the government’s level of support, expressed by the proportion of local government expenditure on agriculture, forestry, and water affairs to GDP. $Traffic_{it}$ indicates the degree of traffic development, expressed in miles per 10,000 people. $\psi_{i}$ represents the individual fixed effect, and $\varepsilon_{it}$ represents the random error term.

The steps are as follows:

Step 1: to test Hypothesis 1, this paper used DIFI as the explanatory variable and DI as the explanatory variable for regression analysis. If the DIFI is significant and the coefficient is greater than zero, then Hypothesis 1 holds.

Step 2: To further test Hypothesis 1, this paper replaced the explanatory variable DIFI, in turn, with six sub-indices, which were DCB, DUD, DSS, Payment, Insurance, and Credit, and the explanatory variable DI, and a detailed analysis was performed (the case of DCB is shown in Formula (2)). The aim is to figure out which parts of the DIFI increase rural incomes. If the variable is significant and the coefficient is greater than zero, then it positively affects the per capita income of disposable income.
(2)$$DI_{it}=\alpha_{0}+\alpha_{1}DCB_{it}+Open_{it}+Edu_{it}+Urban_{it}+Govern_{it}+Traffic_{it}+\psi_{i}+\varepsilon_{it}$$

Step 3: to further test Hypothesis 1, the explanatory variables DI were replaced by four categories of income, which were WI, OI, PI, and TI, and the explanatory variable was DIFI, and a regression analysis was performed (the case of WI is shown in Formula (3)). The aim of this study was to find out which income types of rural residents are positively influenced by DIFI. If the variable is significant and the coefficient is greater than zero, it has a positive effect on the income.
(3)$$WI_{it}=\alpha_{0}+\alpha_{1}DIFI_{it}+Open_{it}+Edu_{it}+Urban_{it}+Govern_{it}+Traffic_{it}+\psi_{i}+\varepsilon_{it}$$

Step 4: to test Hypothesis 2, this paper divided 30 provincial-level administrative units into three regions: the eastern, central, and western regions, and took the total per capita disposable income of rural residents in each region as the explanatory variable, and DIFI as an explanatory variable for heterogeneity analysis. If the DIFI is significant and the coefficient is greater than zero, Hypothesis 2 holds.

In order to test Hypothesis 3, this paper adopted the spatial econometric model; the details are explained in the fifth part.

### 3.3. Data Description

This paper selected data from 30 provincial-level administrative units in China from 2011 to 2018, excluding Tibet, Hong Kong, Macao, and Taiwan. The data come from the China Statistical Yearbook, the China Rural Statistical Yearbook, the China Population & Employment Statistical Yearbook, the National Bureau of Statistics of the People’s Republic of China, and The Peking University Digital Financial Inclusion Index of China (PKU-DFIIC) Report (2011–2018). When the same indicator data were in different databases, the National Bureau of Statistics of the People’s Republic of China data prevailed. In order to unify dimensions and eliminate the effect of heteroscedasticity, this paper takes natural logarithms for all variables. At the same time, to eliminate the effect of price rise, in this paper, we use the 2011-based GDP deflator to deflate all price-based variables. Table 1 reports the descriptive statistics for each variable.

## 4. The Empirical Analysis of Digital Inclusive Finance and the Income of Rural Residents

### 4.1. Digital Inclusive Financial Revenue Effect

Table 2 reports the regression results of the digital inclusive financial index and its six sub-indicators on rural ’”households’ per capita disposable income. Model (1) shows that digital inclusive finance significantly affects farmers’ income under the 1% significance level. At the same time, from the perspective of control variables, the degree of opening to the outside world has no significant negative impact on the increase in rural household income, which may be the reason why rural residents have errors in the analysis of domestic and international information, receiving distortion, and understanding bias, etc., therefore, when the total amount of import and export increases, the income of the population decreases, but the effect is not obvious. The average length of schooling and the level of urbanization have a significant positive effect on the increase in rural household income at the 1% level, which means that with the deepening of rural household education and the acceleration of regional urbanization, the income level of farmers will also increase. This may be because higher levels of education have made it possible for farmers to learn more about increasing their income.

In comparison, higher levels of urbanization have made it easier for farmers to access more financial resources, leading to higher incomes. The government’s expenditure on supporting agriculture and the degree of convenient transportation has no significant positive influence on the increase in farmers’ income, which may be because the government’s financial expenditure has increased and the related financial industry has been supported, thus reducing the cost of farmers’ receiving financial support to increase their income. On the other hand, improved transportation makes it more convenient for farmers to seek financing in the financial industry. It increases their preference for financing in the banking industry, thus increasing their income, but the effect is not obvious.

Models (2)~(4) report the regression results of DCB, DUD, and DSS on rural households per capita disposable income. We can see that under the 1% significance level, the coverage, the depth of use, and the digital degree of these three grading indicators have a significant positive impact on farmers’ income. This may be due to changes in the number of people covered, the number of internet payment accounts, and the number of bank accounts tied to them. First, as digital inclusive financial penetration increases, rural residents can also enjoy the convenience brought about by electronic payment. Second, the increase in the frequency of use, the increase in the number of various financial services, the increase in the number of transactions per capita, the increase in the number of transactions per capita, and the actual use of internet financial services are generally good. Rural residents can also choose financial services that suit their own needs. Third, the degree of digitalization is enhanced, the convenience of using financial services is enhanced, and the cost is reduced. Lower barriers to financial products have made rural residents more willing to participate in related financial products, thus increasing their income. This indicates that the increase in account coverage, depth of use, and level of digital support services have enhanced the convenience of payment and played a significant role in increasing farmers’ income.

Model (5)~(7) reports the regression results of Payment, Insurance, and Credit on rural households per capita disposable income. We can see that the payment business has a significant positive effect on the average per capita disposable income of rural households at the level of 10%, which indicates that the increase in the payment business reflects the increase in rural residents’ incomes. With the development of the internet, electronic platform online consumption and offline electronic payment have come into being. The convenience of payment is improved, increasing the number and amount of personal payments and benefiting farmers’ incomes. The insurance business and the credit business are at 1%, which has a significant positive impact on farmers’ average per capita disposable income. It shows that the increase in insurance business and credit business is beneficial to increasing rural residents’ income. With the development of digital inclusive finance, various kinds of insurance products have emerged one after another, the number and amount of insurance premiums per capita have increased, and the protection that farmers can obtain when they carry out a series of value-added projects, such as farming and planting, has been enhanced. Moreover, farmers need not take on inordinate risk, such as production cuts due to climate and policy factors, in order to increase their income. On the other hand, with the development of digital inclusive finance, financial availability increases, and personal consumption credit and small- and micro-operator credit business increases. Therefore, the disposable income of rural residents also increases. Digital inclusive finance can boost rural incomes by boosting consumption, building up risk tolerance, and increasing access to finance.

### 4.2. The Effect of Digital Inclusive on Different Types of Income

Table 3 reports the regression results of DIFI on the wage income, operating income, property income, and transfer income of rural residents. From Table 3, it can be seen that digital inclusive finance has a significant positive effect on rural residents per capita wage income at the level of 5%, and a significant negative effect on property income. On the other hand, at a 1% significance level, digital inclusive finance has a significant positive impact on operating income and transfer income.

#### 4.2.1. Wage Income

Wage income is mainly composed of the wages of farmers engaged in the main occupation, second occupation, etc. The impact of digital inclusive on rural household wage income is positive, which may be because digital inclusive can encourage entrepreneurship, provide financial support to more people with entrepreneurial ideas, enable more enterprises to be generated, and increase the demand for employment, thus increasing the employment opportunities for farmers, while raising the wage level, thereby increasing the wage income of farmers. However, its influence coefficient is only 0.07, which means that digital inclusive finance has no strong effect on wage income, probably due to the limited employment that digital inclusive finance can provide for farmers; which means that it is difficult for this effect to impact many rural residents.

#### 4.2.2. Operating Income

The operating income is the income that the rural residents derive from the production and operation activities of the family unit, which is mainly composed of self-employed storefronts and buying and selling agricultural products. Digital inclusive finance also has a positive impact on the operating income of farmers, which may be due to the following reasons: firstly, as the degree of digitalization increases, all kinds of financial operations can be carried out more conveniently, and farmers can borrow more conveniently. Thus, the repair, the expansion of self-employed shops, and the increased cash flow can also expand the scale of operations and increase sales revenue. Secondly, the popularization of digital inclusive finance has brought about the convenience of payment and settlement, making e-commerce permeate widely in rural areas. At the same time, it can also enable the selling of agricultural and sideline products on electronic platforms and broaden sales channels to increase the operating income of farmers.

#### 4.2.3. Property Income

Property income includes income from financial products, movables, and real estate owned by rural residents, mainly from investment in wealth management products, property operation, transfer of property use rights, etc. Digital inclusive finance significantly negatively impacts property income at the 5% significance level. There may be several reasons. First, the lack of investment channels and credit resources in rural areas has made the problem of illegal fund-raising serious. Furthermore, the development of digital technology has made the fraudulent methods of illegal organizations more covert and diversified, which exacerbates the problem. Second, in 2011–2018, the investment heat of P2P loans rose continuously, and the number of transactions skyrocketed. However, at the same time, in the absence of policy supervision, the number of cumulative problem platforms accounts for as much as 30% of the total number of P2P platforms, even after the refinement of the regulatory policy also occurred, due to in the phenomenon of default. Furthermore, 2018 saw more frequent lightning, the emergence of missing institutions, and frequent runaway events, disrupting the normal financial order. Many online lending institutions use new business methods, such as e-commerce and exclusive agencies, as fronts to attract funds. They spread fake information and fabricate vague product concepts through online platforms, social platforms, and other online models, promising high returns to lure people. Rural residents’ financial literacy is generally low, and they are easily attracted by propaganda models such as leaflets, media, and promotional meetings. When they lack sufficient knowledge of the fund-raising enterprises and professional analysts who can interpret the information, it is difficult for them to distinguish whether a profit method is legal or not, and they may be confused by “gimmicks”, such as high interest rates and fast returns advocated by lawbreakers. As a result, they are deceived when they invest their funds without knowing the risks of their investments, causing economic losses, resulting in a decline in property income.

#### 4.2.4. Transfer Income

The transfer income includes the transfer income from government and units to individuals, which is the income that rural residents can obtain without paying any reciprocal return, mainly from the retirement pension, unemployment benefits, severance pay, housing provident fund, gifts between families, support and so on. The influence of digital inclusive finance on rural household transfer income is positive, which may be because digital inclusive finance encourages entrepreneurship, the number of self-employed households and private enterprises increases, the number of jobs increases, the number of rural households choosing to go out for work increases the employment rate, and individual security increases, thus increasing farmer household transfer income.

### 4.3. The Effect of Digital Inclusive Finance on Different Regions

Because of the differences in the economic development levels of the east, middle, and west of China, developing digital inclusive finance in different regions may have different effects on farmers’ income. To investigate the heterogeneity of the effect of digital inclusive finance on rural residents’ income increase, 30 provincial administrative units in China were divided into eastern, central and western regions. Table 4 reports the regression results on disposable income. As can be seen from Table 4, digital inclusive finance has a significant income-increasing effect on rural residents in the eastern, central, and western regions under the 1% significance level. It is proven that digital inclusive finance can increase farmers’ incomes. On a regional basis, the effect of digital inclusive finance on revenue growth in the eastern and central regions is the same, with a coefficient of 0.113. In contrast, the impact on the western region is slightly smaller. This may be due to the following reasons: first, the economic development level in eastern and central China is slightly higher than that in western China, thus generating more jobs and better investment targets, while the western region lacks employment opportunities and investment opportunities. Even with the development of digital inclusive finance, most of the funds raised can only be used for consumption, and it is difficult to use them as start-up funds to drive income growth, as this is not sustainable; second, with the corresponding lack of infrastructure in economically backward areas, there are fewer resources, such as broadband, base stations, and electronic devices, so that even with the development of digital inclusive finance, the benefits will not be as high as other regions; third, in economically backward areas, rural residents may have lower financial literacy, with a poor understanding and acceptance of digital inclusive finance, and a lack of the knowledge necessary to understand and use it, so the benefits are greater in regions with high economic levels, where they can promote the local rural resident income growth.

## 5. Further Analysis

The above empirical analysis found that digital inclusive finance can promote an increase in farmers’ incomes through e-commerce and other internet models, so to test Hypothesis 3 in order to ascertain whether there is a spatial effect of digital inclusive finance on farmers’ income, a spatial econometric model was established in this paper for further analysis.

### 5.1. Setting of Spatial Metrology Model

In this paper, we used the spatial autoregressive model (SAR) to investigate further the impact of digital inclusive finance on rural household income:(4)$$DI_{it}=\rho WDI_{it}+\alpha_{0}DIFI_{it}+\alpha_{1}Open_{it}+\alpha_{2}Edu_{it}+\alpha_{3}Urban_{it}+\alpha_{4}Govern_{it}+\alpha_{5}Traffic_{it}+\psi_{i}+\varepsilon_{it}$$

In formula (4), *W* is the space weight matrix, ρ is the autoregressive space coefficient, α is the estimated coefficient of the independent variable, and the other variables have the same meaning as above. If the DIFI is significant and the coefficient is greater than zero, Hypothesis 3 holds.

### 5.2. Empirical Test

In this paper, three kinds of spatial weight matrixes were used: one was the geographical distance weight matrix (W1), which is constructed from the geographical distance between regions; the second was the economic distance weight matrix (W2), which is constructed by the disparity of economic level among regions. The average annual GDP per capita of each province from 2011 to 2018 was used to represent the economic level of each region. The third was the geographical and economic distance nested weight matrix (W3) The matrix formula is: W3 = φW1 + (1 − φ) · W2, φ is between 0 and 1. This matrix is nested W1 and W2, and the dual spatial effects of geographical distance and economic distance are comprehensively considered. To simplify the analysis, the value of φ in this paper is 0.5.

In this paper, we used the Moran’I index to test the spatial correlation of the income of rural residents [41]:(5)$$\mathrm{Moran}’I=\frac{\sum_{i=1}^{n}\sum_{j}^{n}W_{ij}\left(Y_{i}-\bar{Y}\right)\left(Y_{j}-\bar{Y}\right)}{S^{2}\sum_{i=1}^{n}\sum_{j}^{n}W_{ij}}$$

In Formula (5), $Y_{i}$ and $Y_{j}$ represent the income of farmers in *i* and *j* regions, $\bar{Y}$ and $S^{2}$ represent the sample mean and variance. The 2011–2018 Moran’I index for each province is based on the geographical distance matrix, the economic distance matrix, and the nested geographical and economic distance matrix is calculated by the formula, as shown in Table 5. The Moran’I index estimates the average rural population of each province in 2011–2018 based on the geographical distance matrix, the economic distance matrix, and the disposable income matrix.

As can be seen from Table 5, from 2011 to 2018, the Moran’I index of all kinds of rural residents’ disposable income is positive, which shows that under different geographical and economic weights, the growth of farmers’ income in adjacent regions has positive spatial autocorrelation, that is, farmers’ income presents positive correlation in different spatial matrices, has strong spatial dependence and agglomeration effect, and the closer the regions are, the more obvious the mutual influence is. Therefore, this paper selected the spatial autocorrelation (SAR) model to estimate the parameters of the three spatial matrices and the digital inclusive financial index. From Table 6, it can be seen that the spatial autocorrelation ρ of the three spatial weight matrixes is significantly positive, which indicates that the increase in farmers’ income level in this region can promote the increase in farmers’ income in neighboring regions; that is, there is a positive spatial spillover effect. In addition, from three different dimensions, the development of digital inclusive finance has a significant positive impact on the income of rural residents in all regions.

In order to explore the income-increasing effect of developing digital inclusive finance on farmers, this paper decomposed the SAR model effect coefficient. The SAR model’s direct, indirect, and total effects are reported in Table 7. According to Table 7, under the 1% significance level, the direct, indirect, and total effects of digital inclusive finance on local farmers’ income increase are all positive. In terms of direct effects, the development of digital inclusive finance in the region can increase farmers’ income. In regard to indirect effect, the deepening of regional digital inclusive financial development not only enhances the income level of farmers in the region but also promote the income growth of farmers in the neighboring regions; that is, there are regional spillover effects and radiation effects in the growth of rural household income promoted by digital inclusive finance, which confirms Hypothesis 3. This may be because, with the development of digital inclusive finance, the number of new businesses that can be funded increases, the number of jobs increases, the income of farmers in the region increases, and at the same time, rural households in the neighboring regions, attracted by the prospect of employment, also choose to work away from home, thus increasing the income of rural residents in the surrounding areas. The above figures on the financial transfer income of rural residents and the regression results also support this conclusion.

## 6. Conclusions and Countermeasures

Based on the data of Chinese 30 provinces and cities from 2011 to 2018, this paper empirically tested the income-increasing effect of the rural inclusive financial ecological environment on rural residents. First, the development of rural digital inclusive finance can increase the overall income of rural households, which confirms Hypothesis 1, and shows positive effects on the growth of wage income, operating income, and transfer income. Second, digital inclusive finance has a positive income-increasing effect on rural residents in the east, middle and west regions of China, which confirms Hypothesis 2, and compared to China’s western regions, digital inclusive finance is more conducive to the income growth of rural residents in eastern and central China. There is a significant positive spatial correlation between the per capita disposable income of rural residents in different regions of China and the development of digital inclusive finance, and in terms of indirect effect, the development of digital inclusive finance can not only promote an increase in the local farmers’ income, it can also promote an increase in farmers’ incomes in the surrounding areas, and has a positive spillover effect, which is beneficial to the realization of common prosperity and confirms Hypothesis 3.

In conclusion, we believe that inclusive finance with digital technology can help more people, especially in less developed and poorer areas, to access affordable financial services. It is, therefore, effective in reducing poverty and promoting economic growth. At present, the inclusive financial development goals in the world are beginning to attach importance to the digital factor to enhance the efficiency of work. China, as the host of G20 in 2016, actively advocated the development of digital inclusive finance to enhance the role of digital technology in increasing income and poverty alleviation [42].

Based on this, this paper puts forward the following policy recommendations:

Firstly, accelerating the development of digital inclusive finance to promote the income growth of rural residents. Initially, we must improve the construction of digital inclusive financial infrastructure in rural areas, break the restrictions imposed by objective and rigid conditions, increase diversified government and market input, further improve infrastructures, such as payment, clearing, information and communications, continue to expand the coverage of electronic facilities in rural areas [43] in cooperation with mobile communication providers, increase the laying of network facilities in remote areas, use scale to reduce costs, enhance the financial availability of farmers, and boost revenue growth. Then, it is necessary to improve the financial literacy of farmers through the construction of service stations and expansion of the ranks of information workers to increase the means of establishing a comprehensive rural information service system, or to jointly carry out professional knowledge lectures and training with village committees. We should strengthen the education and popularization of digital financial knowledge, understand and correctly use financial tools, and popularize financial knowledge through radio, television, and mobile phone news.

Secondly, optimizing the rural financial ecological environment and paying attention to digital technology development. Digitalization is a key technical tool to enhance financial inclusion in underdeveloped regions. We should expand digital inclusive financial coverage and strengthen digital support to create an inclusive digital inclusive financial system, improving digital capabilities, such as personal payments, microcredit, and insurance, increasing the availability of financial services and products, solving information asymmetry problems, and expanding investment channels and credit resources for rural residents. At the same time, the development of digital technology and other industry integration increases rural residents’ investment opportunities and employment opportunities to help farmers increase production and income.

Thirdly, optimizing the financial regulatory environment to strengthen government supervision and guidance. In China, digital technology is used in a wide range of financial services. It can be used to withdraw money, transfer money, and make payments, through short messages, telephone calls, and the internet. However, there is an inevitable risk of leakage of users’ information. Some financial institutions that sell products such as wealth management and insurance online are themselves at risk. In addition, rural residents generally lack financial knowledge, awareness of self-protection, and the ability to bear risks. Furthermore, with the rapid development of network technology, telecom fraud is more and more rampant, especially in mobile payments, which has brought many obstacles to the development of digital inclusive finance. Therefore, we propose to integrate big data into financial business, establish a sound customer identification system, and actively build a comprehensive management system with prevention in advance, control in the event, and supervision after the event as its core principles. Furthermore, we advocate effective supervision of financial institutions and products to protect the rights and interests of consumers.

## Figures and Tables

**Table 1 ijerph-19-02486-t001:** Descriptive statistics of variables.

Variable	Number of Observations	Mean	Standard Deviation	Minimum	Maximum
DI	240	9.206	0.367	8.271	10.177
WI	240	8.243	0.623	6.690	9.808
OI	240	8.211	0.387	6.776	8.917
PI	240	5.439	0.731	3.691	7.425
TI	240	7.293	0.609	5.891	8.857
DIFI	240	5.073	0.670	2.909	5.934
DCB	240	4.904	0.832	0.673	5.869
DUD	240	5.058	0.644	1.911	5.992
DSS	240	5.392	0.734	2.026	6.117
Payment	240	4.864	0.919	−4.605	5.939
Insurance	240	5.762	1.010	−1.386	6.745
Credit	240	4.645	0.639	0.148	5.493
Open	240	−1.768	0.942	−4.087	0.437
Edu	240	2.294	0.098	2.091	2.607
Urban	240	−0.582	0.204	−1.051	−0.110
Govern	240	−3.692	0.600	−4.850	−2.328
Traffic	240	3.342	0.569	1.639	4.770

**Table 2 ijerph-19-02486-t002:** DIFI and its sub-indicators on the regression results of per capita disposable income.

Variable	Model (1) DIFI	Models (2) DCB	Model (3) DUD	Model (4) DSS	Model (5) Payment	Model (6) Insurance	Model (7) Credit
DIFI	0.139 ***						
	(0.010)						
DCB		0.086 ***					
		(0.016)					
DUD			0.103 ***				
			(0.012)				
DSS				0.090 ***			
				(0.010)			
Payment					0.050 *		
					(0.028)		
Insurance						0.046 ***	
						(0.014)	
Credit							0.056 ***
							(0.016)
Open	−0.019	−0.019	−0.006	−0.011	−0.007	−0.028	0.005
	(0.031)	(0.038)	(0.034)	(0.030)	(0.035)	(0.040)	(0.038)
Edu	1.116 ***	1.485 ***	1.385 ***	1.049 ***	1.614 ***	1.471 ***	1.736 ***
	(0.289)	(0.360)	(0.360)	(0.284)	(0.423)	(0.373)	(0.434)
Urban	1.374 ***	1.547 ***	1.680 ***	1.786 ***	1.913 ***	1.778 ***	1.987 ***
	(0.196)	(0.256)	(0.215)	(0.165)	(0.276)	(0.236)	(0.236)
Govern	0.05	0.086 *	0.080 *	0.087 **	0.095 *	0.117 **	0.114 **
	(0.036)	(0.047)	(0.040)	(0.039)	(0.053)	(0.047)	(0.052)
Traffic	0.12	0.147	0.185	0.215 *	0.221 *	0.289 **	0.225
	(0.099)	(0.128)	(0.110)	(0.114)	(0.129)	(0.137)	(0.147)
Constant term	6.489 ***	6.070 ***	6.149 ***	6.936 ***	5.972 ***	6.016 ***	5.796 ***
	(0.774)	(0.943)	(0.958)	(0.668)	(1.034)	(0.985)	(1.122)
Individual fixation effect	Yes	Yes	Yes	Yes	Yes	Yes	Yes
Observations	240	240	240	240	240	240	240
adj. R^2^	0.943	0.925	0.924	0.943	0.916	0.92	0.904

Note: ***, ** and * are significant at 1%, 5% and 10% significance levels, respectively. Robust standard errors of estimated coefficients are shown in parentheses.

**Table 3 ijerph-19-02486-t003:** Digital inclusive regression results for the four categories of revenue.

Variable	Model (8) Wage Income	Model (9) Operating Income	Model (10) Property Income	Model (11) Transfer Income
DIFI	0.070 **	0.116 ***	−0.179 **	0.491 ***
	(0.026)	(0.017)	(0.071)	(0.075)
Open	−0.125 *	−0.082 **	0.203	0.324 ***
	(0.062)	(0.035)	(0.127)	(0.066)
Edu	1.630 **	0.459	0.988	2.414 **
	(0.595)	(0.407)	(1.274)	(1.075)
Urban	1.453 ***	0.717 **	2.784 ***	2.276 ***
	(0.337)	(0.286)	(0.844)	(0.802)
Govern	0.186 **	−0.068	0.256	0.013
	(0.068)	(0.053)	(0.172)	(0.175)
Traffic	0.148	−0.006	1.134 **	0.574
	(0.157)	(0.138)	(0.493)	(0.452)
Constant term	4.964 ***	6.610 ***	3.216	−0.705
	(1.560)	(1.333)	(2.983)	(3.446)
Individual fixation effect	Yes	Yes	Yes	Yes
Observations	240	240	240	240
adj. R^2^	0.798	0.815	0.376	0.806

Note: ***, ** and * are significant at 1%, 5% and 10% significance levels, respectively. Robust standard errors of estimated coefficients are shown in parentheses.

**Table 4 ijerph-19-02486-t004:** DIFI’s regression results on per capita disposable income of rural households in eastern, central, and western China.

Variables	Eastern	Central	Western
DIFI	0.113 ***	0.113 ***	0.110 ***
	(0.021)	(0.012)	(0.011)
Open	−0.271 ***	−0.096 **	0.055 *
	(0.036)	(0.036)	(0.026)
Edu	1.825 ***	−0.273	0.641 **
	(0.336)	(0.215)	(0.260)
Urban	0.674	1.919 ***	1.730 ***
	(0.382)	(0.175)	(0.210)
Govern	0.048	0.033	0.111 **
	(0.064)	(0.041)	(0.044)
Traffic	0.195	0.19	0.133
	(0.125)	(0.231)	(0.158)
Constant term	4.299 ***	9.696 ***	8.182 ***
	(0.735)	(1.091)	(0.992)
Individual fixation effect	Yes	Yes	Yes
Observations	88	64	88
adj. R^2^	0.952	0.965	0.965

Note: ***, ** and * are significant at 1%, 5% and 10% significance levels, respectively. Robust standard errors of estimated coefficients are shown in parentheses.

**Table 5 ijerph-19-02486-t005:** Moran’I index for disposable income.

		2011	2012	2013	2014	2015	2016	2017	2018
W1	Moran’I	0.182	0.182	0.190	0.190	0.187	0.183	0.181	0.180
	Z	6.162	6.171	6.436	6.432	6.351	6.253	6.190	6.168
	*p*-value	0.000	0.000	0.000	0.000	0.000	0.000	0.000	0.000
W2	Moran’I	0.393	0.395	0.386	0.387	0.391	0.392	0.393	0.393
	Z	5.533	5.558	5.475	5.488	5.535	5.559	5.574	5.588
	*p*-value	0.000	0.000	0.000	0.000	0.000	0.000	0.000	0.000
W3	Moran’I	0.262	0.263	0.268	0.268	0.269	0.267	0.266	0.265
	Z	5.294	5.309	5.428	5.440	5.451	5.427	5.417	5.407
	*p*-value	0.000	0.000	0.000	0.000	0.000	0.000	0.000	0.000

Note: Z is the Z value of the Moran’I index after normalization, and *p*-Value is the probability value corresponding to the Z value.

**Table 6 ijerph-19-02486-t006:** Baseline regression results for SAR models.

Variable	W1	W2	W3
DIFI	0.023 ***	0.020 ***	0.020 ***
	(0.006)	(0.007)	(0.007)
Open	0.017	0.017	0.018
	(0.012)	(0.015)	(0.014)
Edu	−0.123	−0.093	−0.142
	(0.093)	(0.109)	(0.101)
Urban	0.499 ***	0.455 ***	0.473 ***
	(0.118)	(0.161)	(0.133)
Govern	−0.019	−0.035 *	−0.029
	(0.018)	(0.020)	(0.020)
Traffic	0.055	0.058	0.063
	(0.046)	(0.059)	(0.053)
ρ	0.813 ***	0.816 ***	0.817 ***
	(0.039)	(0.045)	(0.039)
Observations	240	240	240
adj. R^2^	0.971	0.970	0.972

Note: *** and * are significant at 1% and 10% significance levels, respectively. Robust standard errors of estimated coefficients are shown in parentheses.

**Table 7 ijerph-19-02486-t007:** Direct effects, indirect effects, and total effects of SAR models.

Variable	Direct Effects			Indirect Effects			Total Effects		
	W1	W2	W3	W1	W2	W3	W1	W2	W3
DIFI	0.026 ***	0.025 ***	0.023 ***	0.098 ***	0.088 ***	0.087 ***	0.124 ***	0.113 ***	0.110 ***
	(0.007)	(0.008)	(0.007)	(0.030)	(0.029)	(0.028)	(0.034)	(0.035)	(0.034)
Open	0.019	0.02	0.02	0.078	0.084	0.085	0.097	0.103	0.105
	(0.013)	(0.019)	(0.016)	(0.067)	(0.097)	(0.084)	(0.079)	(0.115)	(0.099)
Edu	−0.132	−0.101	−0.156	−0.532	−0.367	−0.606	−0.664	−0.468	−0.762
	(0.103)	(0.129)	(0.114)	(0.490)	(0.548)	(0.520)	(0.585)	(0.668)	(0.625)
Urban	0.565 ***	0.541 ***	0.547 ***	2.104 ***	1.896 ***	2.024 ***	2.669 ***	2.436 ***	2.571 ***
	(0.116)	(0.168)	(0.136)	(0.423)	(0.510)	(0.484)	(0.472)	(0.621)	(0.567)
Govern	−0.022	−0.042 *	−0.033	−0.091	−0.16	−0.134	−0.113	−0.202	−0.167
	(0.020)	(0.025)	(0.023)	(0.095)	(0.120)	(0.111)	(0.114)	(0.142)	(0.133)
Traffic	0.065	0.074	0.077	0.257	0.299	0.305	0.322	0.373	0.382
	(0.051)	(0.073)	(0.062)	(0.221)	(0.322)	(0.264)	(0.270)	(0.390)	(0.323)

Note: *** and * are significant at 1% and 10% significance levels, respectively. Robust standard errors of estimated coefficients are shown in parentheses.

## Data Availability

Publicly available datasets were analyzed in this study. These data can be found here: http://www.stats.gov.cn; https://navi.cnki.net/knavi (accessed on 20 January 2022).

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
