# Peer review of "Research on the Effect of Rural Inclusive Financial Ecological Environment on Rural Household Income in China"

_ijerph, 2022, doi:10.3390/ijerph19042486_

Round 1
Reviewer 1 Report
The topic chosen by the authors is analyzed at both theoretical and practical levels. The topic is interesting and original. Despite this, there are some aspects that the authors should pay attention to:
- The introduction part must be improved. It is not clear enough, what scientific problem are you going to solve? What is the novelty of your investigation compared with others?
- All the statements in the text must be proved, so please indicate the literature sources of all statistical data you have mentioned in the Introduction part.
- Please indicate full literature sources according to the Journal‘s requirements. It is not clear when you write, for example, Andrew Lawson, Heidi ant etc. Usually, it is written „Lawson (2017) <....>.
- What do numbers mean? For example, „easing financial exclusion11“? If it is a quotation, it must match the requirements of the Journal. Usually, it is written according to the APA style.
- How will the hypothesis be tested? It would be good to clarify the „3.1. Theoretical mechanism“ chapter.
- In „Data description“ you are going to use data only till 2018. Now it is 2022. Wasn‘t it possible to refresh the data?
- Why the description of table 2 is in page 5, but the table is only in page 7? Couldn‘t they be close to each other?
- Would it be possible to clarify, what are models you use? For example, it is written „From model (2) ~ (4) <...>“; „From the model (5) ~ (7) <...>“, but only on page 7 it is shown in the table à it would be recommended to describe it before in the methodology part.
- The Reference list must be formalized in accordance with the requirements of the Journal. Not all references are made like this.
RECOMMENDATION: REVISIONS REQUIRED
Author Response
Response to Reviewer 1 Comments
Point 1: The introduction part must be improved. It is not clear enough, what scientific problem are you going to solve? What is the novelty of your investigation compared with others?
Response 1: The introduction has been improved by adding the description of the research contents, research significance, and innovation points.
The widening of poverty and the income gap has become one of the serious challenges facing the development of the world today, which has attracted the attention of the international community, especially the developing countries[1,2].
In September 2016, China proposed at the G20 Hangzhou summit the formulation of the G20 senior principles on digital inclusive finance, advocating the use of digital technologies to support the development of inclusive finance construction of a digital financial infrastructure ecosystem.
China is the largest developing country globally, and the regional development is quite different. Studying the revenue-increasing effect of Chinese digital inclusive finance has important reference significance for policymakers to design and implement poverty reduction strategies, reducing the incidence of poverty and income inequality[5].
Point 2: All the statements in the text must be proved, so please indicate the literature sources of all statistical data you have mentioned in the Introduction part.
Response 2: All of the statistics mentioned in the introduction are from the National Bureau of Statistics of the People’s Republic of China and explained accordingly.
The Chinese government has made a series of remarkable achievements since 2013, according to the National Bureau of Statistics of the People’s Republic of China.
Point 3: Please indicate full literature sources according to the Journal‘s requirements. It is not clear when you write, for example, Andrew Lawson, Heidi ant, etc. Usually, it is written „Lawson (2017) <....>.
Response 3: According to the magazine’s request, we have indicated the complete literature source. For example:
Galak argues that digital finance will allow a rapid expansion of financial services to reach more people[37].
Point 4: What do numbers mean? For example, „easing financial exclusion11“? If it is a quotation, it must match the requirements of the Journal. Usually, it is written according to the APA style.
Response 4: The number in the literature review is the reference number of the last document, which has been improved according to the new format. For example:
Exchange rates, gross domestic product, and technology can boost rural incomes, but interest rates and rural land prices are inversely related to rural incomes[16].
Point 5: How will the hypothesis be tested? It would be good to clarify the „3.1. Theoretical mechanism“ chapter.
Response 5: Ways to test hypotheses have been added to modelling.
In order to test hypotheses 1 and 2 and study the effect of digital inclusive finance on the income of rural residents, the following individual fixed effect model is constructed.
In order to test hypothesis 3, this paper adopts the spatial econometric model; the details are explained in the fifth part.
Point 6: In „Data description“ you are going to use data only till 2018. Now it is 2022. Wasn‘t it possible to refresh the data?
Response 6: For the time being, only 2018 data will be available, as some of the data needed for 2020 are still not publicly disclosed.
Point 7: Why the description of table 2 is in page 5, but the table is only in page 7? Couldn‘t they be close to each other?
Response 7: Table 2 has been repositioned.
Point 8: Would it be possible to clarify, what are models you use? For example, it is written „From model (2) ~ (4) <...>“; „From the model (5) ~ (7) <...>“, but only on page 7 it is shown in the table à it would be recommended to describe it before in the methodology part.
Response 8: The econometric model used in this paper has been added with a detailed description in the modelling.
The steps are as follows:
Step 1: to test hypothesis 1, this paper uses DIFI as the explanatory variable and DI as the explanatory variable for regression analysis. If the DIFI is significant and the coefficient is greater than zero, then hypothesis 1 holds.
Step 2: To further test hypothesis 1, this paper replaces the explanatory variable DIFI, in turn, with six sub-index, which is DCB, DUD, DSS, Payment, Insurance, and Credit, and the explanatory variable DI, and perform a detailed analysis (in the case of DCB, as shown in formula 2). The aim is to figure out which parts of the DIFI increase rural incomes. If the variable is significant and the coefficient is greater than zero, then it has a positive effect on the per capita income of disposable income.
|
|
(2) |
Step 3: to further test hypothesis 1, the explanatory variables DI were replaced by four categories of income which is WI, OI, PI, TI, and the explanatory variables is DIFI, and perform a regression analysis (in the case of WI, as shown in formula 3). The aim of this study is to find out which income types of rural residents are positively influenced by DIFI. If the variable is significant and the coefficient is greater than zero, it has a positive effect on the income.
|
|
(3) |
Step 4: to test hypothesis 2, this paper divides 30 provincial-level administrative units into three regions: the eastern, central, and western regions, and takes the total per capita disposable income of rural residents in each region as the explanatory variable, DIFI as an explanatory variable for heterogeneity analysis. If the DIFI is significant and the coefficient is greater than zero, hypothesis 2 holds.
Point 9: The Reference list must be formalized in accordance with the requirements of the Journal. Not all references are made like this.
Response 9: The format of references has been readjusted as required. For example:
Jin, D. The Inclusive Finance Have Effects on Alleviating Poverty. Open Journal of Social Sciences 2017, 05, 233, doi:10.4236/jss.2017.53021.

Reviewer 2 Report
The article proposal meets the scientific criteria for publication in your journal.
It presents an incursion in the stage of relatively comprehensive knowledge on the subject treated in the article; perhaps an increase in the number of bibliographic references allocated to this research topic would be advisable.
For the application part, I appreciate the quantitative research component that offers consistency and authenticity to this article proposal.
Author Response
Thank you for your recognition of our research results!
Reviewer 3 Report
This study the effect of digital inclusive finance on farmers’ income. I listed my suggestions below in detail for improvement.
Explain how the econometric model was arrived at. Its essential for all readers to understand your study. -Is it a standard equation or developed my the authors.
Why 2011-based GDP deflator to deflate all price-based variables was used? – which is ten years from the original data considered.
Line287 & 312– Table 3, Table 4– change it.
Can this study be applied to developing nations?
What’s author's opinion on digital inclusive finance on farmers in poor countries.
Cite more relevant references from the journal.
Author Response
Response to Reviewer 3 Comments
Point 1: Explain how the econometric model was arrived at. Its essential for all readers to understand your study. -Is it a standard equation or developed my the authors.
Response 1: As to how the econometric model is obtained, this paper has added a detailed description in the modeling.
In order to test hypotheses 1 and 2 and study the effect of digital inclusive finance on the income of rural residents, the following individual fixed effect model is constructed.
The steps are as follows:
Step 1: to test hypothesis 1, this paper uses DIFI as the explanatory variable and DI as the explanatory variable for regression analysis. If the DIFI is significant and the coefficient is greater than zero, then hypothesis 1 holds.
Step 2: To further test hypothesis 1, this paper replaces the explanatory variable DIFI, in turn, with six sub-index, which is DCB, DUD, DSS, Payment, Insurance, and Credit, and the explanatory variable DI, and perform a detailed analysis (in the case of DCB, as shown in formula 2). The aim is to figure out which parts of the DIFI increase rural incomes. If the variable is significant and the coefficient is greater than zero, then it has a positive effect on the per capita income of disposable income.
|
|
(2) |
Step 3: to further test hypothesis 1, the explanatory variables DI were replaced by four categories of income which is WI, OI, PI, TI, and the explanatory variables is DIFI, and perform a regression analysis (in the case of WI, as shown in formula 3). The aim of this study is to find out which income types of rural residents are positively influenced by DIFI. If the variable is significant and the coefficient is greater than zero, it has a positive effect on the income.
|
|
(3) |
Step 4: to test hypothesis 2, this paper divides 30 provincial-level administrative units into three regions: the eastern, central, and western regions, and takes the total per capita disposable income of rural residents in each region as the explanatory variable, DIFI as an explanatory variable for heterogeneity analysis. If the DIFI is significant and the coefficient is greater than zero, hypothesis 2 holds.
In order to test hypothesis 3, this paper adopts the spatial econometric model; the details are explained in the fifth part.
Point 2: Why 2011-based GDP deflator to deflate all price-based variables was used? – which is ten years from the original data considered.
Response 2: 2011 was the first year of public disclosure for the Beijing University Digital Inclusive Financial Index (DIFI), and no earlier data were available. In order to ensure the consistency of the base period of various price data, this paper takes 2011 as the base period.
Point 3: Line287 & 312– Table 3, Table 4– change it.
Response 3: The order of tables 3 and 4 has been changed and added explanatory text.
Point 4: Can this study be applied to developing nations?
Response 4: Whether the study can be applied to all developing countries remains to be seen, but studies in China and other countries such as India and Kenya confirm that the development of digital inclusive finance can help rural residents lift themselves out of poverty and increase their incomes, we believe that the future development of this research is bright. The introduction of this paper also adds relevant descriptions.
Point 5: What’s author's opinion on digital inclusive finance on farmers in poor countries.
Response 5: Our view on digital inclusive finance for farmers in poor countries is that inclusive finance with digital technology can help more people, especially those in less developed and poor areas, to access affordable financial services. It is, therefore, more effective in reducing poverty and promoting economic growth. In the conclusion part of this paper, a description is also added.
Point 6: Cite more relevant references from the journal.
Response 6: We add 11 relevant pieces of literature. See the literature review for details.

Reviewer 4 Report
The paper proposes to speed up the development of digital inclusive finance, optimize the rural financial ecological environment, strengthen government supervision and other recommendations, further enhance farmers' income, and achieve common prosperity. This paper uses a spatial econometric model and finds that promoting the development of local digital inclusive finance will enhance the income level of local farmers and increase the income of neighbouring farmers.
I cannot support the progression of this paper because it is at too early a stage. The paper needs more work before it can be considered for publication. The paper itself I believe has been submitted too early and would benefit from another round of revisions and drafting.
The paper needs to address a number of key issues:
- The contribution of the paper to the literature is unclear and its importance in relation to what has gone before is not properly articulated. What does the paper add that is new and makes it worth publishing in a leading journal. This is currently unclear.
- The paper is not structured properly yet and could do with another round of major edits. Conclusion and Countermeasures are not proportional to the arguments based on empirical data and should be described in more detail here.
- The literature review is not extensive, and does not position the paper well enough yet. A lot of it is descriptive of what is in the papers and this makes the discussion fragmented rather than cumulative, to identify a key gap in our knowledge or a problem that the paper is addressing. This needs work.
- The selection of the case and its role is under specified. I believe that China on its own is an interesting enough case to look at, but I expected to see a justification of why its useful to explore and what kind of case it is. What lessons can we learn? The paper generalises from the China case, but the readers is not informed about how specific the case is and whether that generalisation is appropriate.
My suggestions for the authors would be to model their paper more closely on existing papers in terms of content. There are many key features of a paper that I expected to find in this one that were missing.
For example, I cannot agree with the unsupported statement that “Chinese scholars first put forward the concept of the financial ecological environment”. The authors omit Traditional Financial Theory and do not show any link or relationship to this theory. They do not show the genesis and evolution of this concept made by "Chinese Scholars".
Elsewhere, the sentence “We should strengthen the access mechanism of the financial market, build an effective digital inclusive financial supervision system, and reduce the occurrence of illegal fund-raising cases in rural areas” does not meet the scientific standards for a recommendation in a scientific journal.
My overall impression of the paper is that it has been submitted too early and needs more work to get it ready for peer review. I am sure there is a lot of interest in the data and the case, but this is not yet easy to see for the reader. I wish the authors good luck in improving the paper.
Author Response
Response to Reviewer 4 Comments
Point 1: The contribution of the paper to the literature is unclear and its importance in relation to what has gone before is not properly articulated. What does the paper add that is new and makes it worth publishing in a leading journal. This is currently unclear.
Response 1: This paper’s significance, innovation and contribution have been expounded in the literature review.
By combing the previous relevant literature, we find that the optimization of financial ecological environment, the development of inclusive finance, and digital inclusive finance positively affect farmers’ income. With the support of digital technology, the coverage of rural financial services has been improved, making up for the lack of inclusive finance. But many scholars put forward this view based more on the inclusive financial perspective to analyze rural financial services that need to use digital technology to solve the problems of traditional inclusive finance, from the digital point of view to study the current situation of rural finance is still relatively small. Therefore, this paper focuses on the impact of digital inclusive finance on rural income to explore whether the development of digital inclusive finance can further promote rural income growth and in what form to promote its growth. In this paper, the research contents are divided as follows: first, we study the effect of digital inclusive finance aggregate index on farmers’ income. Second, to study the effect of six sub-indicators on farmers’ income. The third is to divide the farmers’ income into four categories and then study the influence of the total index of digital inclusive finance on their sub-income. Finally, the spatial metrological model is used for further analysis.
Point 2: The paper is not structured properly yet and could do with another round of major edits. Conclusion and Countermeasures are not proportional to the arguments based on empirical data and should be described in more detail here.
Response 2: We reorganize the structure of this paper and revise some policy suggestions. See the conclusion and countermeasures section for details.
Based on this, this paper puts forward the following policy recommendations:
Firstly, accelerate the development of digital inclusive finance to promote the income growth of rural residents. First, we will improve the construction of digital inclusive financial infrastructure in rural areas, break the restrictions imposed by objective and rigid conditions, increase diversified government and market input, and further improve infrastructures such as payment and clearing, and information and communications, continue to expand the coverage of electronic facilities in rural areas[44], in cooperation with mobile communication providers, increase the laying of network facilities in remote areas, use scale to reduce costs and enhance the financial availability of farmers, boost revenue growth. Second, to improve the financial literacy of farmers, through the construction of service stations and the ranks of information workers, to increase the means to establish a comprehensive rural information service system, or to jointly carry out professional knowledge lectures and training with village committees, we should strengthen the education and popularization of digital financial knowledge, understand and correctly use financial tools, and popularize financial knowledge through radio, television, and mobile phone news.
Secondly, optimize the rural financial ecological environment and pay attention to digital technology development. Digitalization is a key technical tool to enhance financial inclusion in underdeveloped regions. We should expand digital inclusive financial coverage and strengthen digital support to create an inclusive digital inclusive financial system, improving digital capabilities such as personal payments, microcredit, and insurance, increasing the availability of financial services and products, solving Information asymmetry problems, expanding investment channels and credit resources for rural residents. At the same time, the development of digital technology and other industry integration increases rural residents’ investment opportunities and employment opportunities to help farmers increase production and income.
Thirdly, optimize the financial regulatory environment strengthen the role of government supervision and guidance. In China, digital technology is used in a wide range of financial services, and can be used to withdraw money, transfer money and make payments through short messages, telephone calls, and the internet. Still, there is an inevitable risk of users’ information leakage, and some financial institutions sell products such as wealth management and insurance online that are themselves at risk. In addition, rural residents generally lack financial knowledge, awareness of self-protection, and the ability to bear risks. And now, with the rapid development of network technology, telecom fraud is more and more rampant, especially in mobile payments, which has brought many obstacles to the development of digital inclusive finance. Therefore, we propose to integrate big data into the financial business, establish a sound customer identification system, and actively build a comprehensive management system with prevention in advance, control in the event, and supervision after the event as the core, effective supervision of financial institutions and products to protect the rights and interests of consumers.
Point 3: The literature review is not extensive, and does not position the paper well enough yet. A lot of it is descriptive of what is in the papers and this makes the discussion fragmented rather than cumulative, to identify a key gap in our knowledge or a problem that the paper is addressing. This needs work.
Response 3: We revise the way of expression of language in the literature review to make the expression more logical. For example:
In 1935, the British ecologist Tessler first put forward the concept of ecosystem. In the following decades, financial scholars have been trying to develop the theory of the financial ecosystem. The term “financial ecology” comes from “optimizing financial ecology” put forward by Xiaochuan Zhou in 2004; the two basic concepts of financial ecological subject and financial ecological environment have evolved gradually. Nowadays, the new theory of the financial ecosystem has become more and more perfect.
Point 4: The selection of the case and its role is under specified. I believe that China on its own is an interesting enough case to look at, but I expected to see a justification of why its useful to explore and what kind of case it is. What lessons can we learn? The paper generalises from the China case, but the readers is not informed about how specific the case is and whether that generalisation is appropriate.
Response 4: In the introduction and conclusion of this paper, the importance of the case study is increased.
The widening of poverty and the income gap has become one of the serious challenges facing the development of the world today, which has attracted the attention of the international community, especially the developing countries[1,2].
China is the largest developing country globally, and the regional development is quite different. Studying the revenue-increasing effect of Chinese digital inclusive finance has important reference significance for policy makers to design and implement poverty reduction strategies, reducing the incidence of poverty and income inequality[5].
In conclusion, we believe that inclusive finance with digital technology can help more people, especially in less developed and poorer areas, access affordable financial services. It is, therefore, more effective in reducing poverty and promoting economic growth. At present, the inclusive financial development goals in the world are beginning to attach importance to the digital factor to enhance the efficiency of the work. As the host of the G20 in 2016, China has actively advocated the development of digital inclusive fi-nance to enhance the role of digital technology in increasing income and poverty alle-viation[43].

Round 2
Reviewer 4 Report
Accept in its current form